# OpenReview forum: "Gestalt Generalized Category Discovery"
_ICLR.cc/2026/Conference — Submitted to ICLR 2026_

### Official Review · Reviewer_1QQr · 2025-10-25

**Soundness:** 2
**Presentation:** 2
**Contribution:** 2
**Rating:** 2
**Confidence:** 3

**Summary:**

The paper proposes three feature refinements, inspired by psychological principles, to enhance generalized category discovery by making features richer and better structured. The refinement methods take dense features as input, along with a set of slots—referred to as hyper-relation slots—and output a pooled version of the dense features that follow the psychological factors of proximity, similarity, and continuity. Finally, the pooled dense features and the CLS token are combined and used as input to GCD frameworks. The method demonstrates compatibility with several established state-of-the-art approaches on coarse-grained and fine-grained benchmarks.

**Strengths:**

1 - The method is plug and play, meaning that it can be added to existing methods and improve their performance and evidenced by Table1 and Table 2.

2 - The method is lightweight and only adds slightly higher training compute.

3 - The idea that CLS token is not ideal alone and needs to be augmented by some complementary information that includes the scene structure is interesting and convincing.

**Weaknesses:**

1 - While I agree with the identified issue regarding the CLS token, I do not see a major benefit in the proposed solutions. The paper first introduces relational supports and uses them to pool dense features, but then argues that this alone is insufficient and introduces correction biases. One such bias involves incorporating a similarity matrix at the dense level; however, this already exists implicitly within the CLS token, and I do not see any clear advantage in reintroducing it. Furthermore, the proximity operator aims to make the features of immediate neighbors more similar, which could effectively average out patches depicting different semantics—potentially harming the representation quality. This limitation is also reflected in the ablation study (Table 3), where the contribution of each component is marginal, around 0.5%.

2 - I find the structure of the paper difficult to follow. A solution is first proposed for a problem, but then it is argued that this very solution introduces another issue, leading to additional proposed fixes. It is also unclear whether the slots are trainable or not. Moreover, since the transformer layer itself can be viewed as a message-passing operation, it is not clear how adding another layer of message passing on top of an existing stack of pretrained transformer layers can effectively address the problem introduced by that same architecture.

3 - The ablation studies in Table 4 require more explanation regarding the baseline. HyperGCD is introduced abruptly without any reference or definition. I assume it refers to the final model used for the ablation studies, but its relation to GesGCD should be clarified.

4 - An ablation study on the number of slots should be included, as this factor could have a significant impact on performance.

5 - More recent models, such as DINOv2 and DINOv3, may have already addressed the identified problem to a greater extent. Results based on these backbones should be included to better demonstrate the generality of the proposed method.

6 - The authors claim that the code is available; however, it is only presented as pseudo-code and is not runnable.

**Questions:**

1 - I kindly ask the authors to clarify the ablation studies. For instance, please explain what HyperGCD refers to and include results based on DINOv2 (and possibly DINOv3) backbones to demonstrate the generality of the proposed method.

2 - I also encourage the authors to include an ablation study on the number of slots, clarifying whether these slots are trainable or fixed random vectors.

3 - As I explained earlier, I may have misunderstood parts of the method due to the paper’s structure and presentation. I encourage the authors to improve the clarity of the overall framework, particularly the relation between the proposed refinements, the role of hyper-relation slots, and their integration into the GCD pipeline.

---

### Official Review · Reviewer_Ga9P · 2025-10-26

**Soundness:** 3
**Presentation:** 2
**Contribution:** 2
**Rating:** 2
**Confidence:** 4

**Summary:**

This paper proposes GesGCD, a GCD framework inspired by cognitive science that inserts a hyper-relation construction stage between backbone and classifier to group tokens before category decisions.

**Strengths:**

- Cognitive science motivation provides interesting perspective on GCD
- Addresses a real issue that tokens are often treated atomically in current pipelines
- The goal of aligning machine and human discovery processes is worthwhile
- Claims orthogonality to existing objectives, suggesting potential broad applicability

**Weaknesses:**

- The submission is incomplete, cutting off before presenting the full Gestalt calibration mechanism, experimental results, and ablation studies. This prevents proper evaluation of the core contributions.

- The hyper-relation construction (Eq. 2-4) resembles standard attention mechanisms with learnable queries and keys. The distinction from existing soft clustering or slot attention methods is unclear. What specifically makes this "hyper-relational" versus multi-head attention with slot-based aggregation?

- The cognitive science framing, while appealing, lacks rigorous grounding. The mixture model in Section 2.1 introduces factors {sm} but their relationship to learned hyper-relation slots D is hand-wavy. How does Equation 4's softmax normalization actually implement "competitive grouping" from human perception?

- The Gestalt calibration mechanism is promised but not delivered. How are proximity, similarity, and continuity actually computed and integrated? Are they differentiable? Do they add trainable parameters? The claim of "parameter-free" calibration conflicts with the learnable Wd in Equation 2.

- Missing critical experimental details: which GCD methods were tested, what datasets, what are the actual performance numbers, computational costs, and ablations isolating hyper-relation vs Gestalt components?

- The terminology is unnecessarily heavy. "Anti-atomization" and "hyper-relation" don't have established meanings. Simpler terms like "token grouping" or "relational aggregation" would improve clarity.

- Figure 2 shows components but doesn't explain information flow or how Gestalt calibration interacts with hyper-relation construction. The covariance matrix appearance is unexplained.

**Questions:**

1. How exactly do you implement proximity, similarity, and continuity in the Gestalt calibration? Are they computed from spatial positions, feature distances, or both?

2. Can you provide ablation studies showing: (a) hyper-relation alone, (b) each Gestalt principle individually, (c) the full system?

3. What is the computational overhead? How does M (number of slots) affect performance and cost?

4. You claim the approach is "plug-and-play" with existing GCD frameworks. Can you show results plugging into multiple different base methods (e.g., GCD, ORCA, SimGCD)?

5. The column-stochastic constraint in Eq. 4 normalizes across tokens for each slot. Why not row-stochastic (normalize across slots for each token) as in typical attention?

6. How do you determine M? Is it related to the number of classes, or is it a separate hyperparameter?

---

### Official Review · Reviewer_65CL · 2025-10-29

**Soundness:** 3
**Presentation:** 2
**Contribution:** 3
**Rating:** 6
**Confidence:** 2

**Summary:**

GesGCD introduces a novel cognition-inspired pipeline for Generalized Category Discovery by adding two key stages:
 (1) a Hyper-Relation Construction layer inserted between the feature backbone and the classifier, which groups tokenized image features into higher-order relations rather than treating each token in isolation; and
(2) a Gestalt Psychology Calibration module that parameter-free injects human-like perceptual grouping principles, proximity, similarity, and continuity, into those relations, encouraging the model to cluster features that are spatially or conceptually coherent.
This "grouping-before-induction" strategy is directly inspired by human cognition and requires no additional supervision signals.
The proposed components are plug-and-play (compatible with existing GCD architectures without modifying their objectives) and computationally efficient, incurring only marginal overhead in model size and runtime. Empirically, GesGCD achieves consistently higher accuracy on known, novel, and all classes across multiple benchmarks (spanning fine-grained datasets like CUB and Cars to coarse-grained ones like CIFAR and ImageNet). The approach not only improves quantitative performance but also yields more interpretable representations, evidenced by intuitive visualizations of grouped features and focused attention maps aligning with meaningful object parts.

**Strengths:**

1. Cognitive Science Integration: Introduces a novel integration of Gestalt perceptual grouping theory into a machine learning pipeline, aligning the model’s behavior with human-like grouping principles and offering a fresh perspective on category discovery.
2. Plug-and-Play Architecture: The proposed hyper-relation + Gestalt module is architecturally plug-and-play, fitting between backbone and head without altering their internals. It is compatible with different networks and objectives, making it easy to adopt in existing GCD frameworks.
3. Good Empirical Performance: Demonstrates consistent accuracy improvements on both fine-grained and coarse-grained benchmarks (improving known, novel, and overall class accuracy in all cases). Results show clear gains when GesGCD is added to various baseline methods, underscoring its general effectiveness.
4. Lightweight and Efficient: The method is computationally lightweight with minimal overhead – only a negligible increase in parameters and FLOPs, and virtually unchanged training/inference time. This means the benefits come at almost no cost, an important strength for practical use.

**Weaknesses:**

1. Design Hyperparameters Not Explored: While the Gestalt calibration is largely parameter-free, the approach likely involves some design choices (e.g., how proximity or continuity are quantified). The sensitivity of these settings is not deeply analyzed. An isolated exploration of each Gestalt factor’s effect (beyond the ablation removing it entirely) or any thresholds used would be useful to understand the method’s robustness and optimal configuration fully.
2. Marginal Gains on Coarse Tasks: In some coarse-grained benchmarks, the performance gains, though positive, appear relatively small or incremental. This suggests that when categories are very distinct (e.g., CIFAR-10 classes), the added grouping yields less dramatic benefit. The paper could discuss these cases further to clarify in which scenarios the approach has the most impact and ensure that even modest improvements are statistically significant.
3. This paper is not written in full 9 pages.
4. The experiment datasets do not seem complex, so we do not know the real power of the method proposed in this paper.

**Questions:**

1. To test the cross-scheme and cross-model compatibility on coarse-grained Datasets, you only use CIFAR-10 and ImageNet-100. It is strange in 2025 that you still use this simple and small dataset to test, as there are so many image datasets. Can you provide test results on other datasets?
2. Did you observe any failure modes where the grouping mechanism misbehaves (for instance, over-grouping distinct classes together, or under-grouping where meaningful connections aren’t formed)? An analysis of such cases would be valuable.

---

### Official Review · Reviewer_nW2A · 2025-10-31

**Soundness:** 1
**Presentation:** 1
**Contribution:** 1
**Rating:** 0
**Confidence:** 4

**Summary:**

This paper takes inspiration from gestalt psychology to improve a known application of generalized category discover (GCD) -- a class of semi-supervised learning problem where images from the unlabeled set may belong to novel categories that have not been observed in the labeled set. Often, GCD approaches employ 3 stages: (1) feature extraction, (2) self-supervised contrastive learning, (3) clustering. This paper takes aim at the second stage, by introducing psychologically inspired inductive biases.

**Strengths:**

This paper connects ideas from gestalt psychology (low-level perceptual organization) to a problem in semi-supervised learning. How I see it, the paper aims to bring known inductive biases (closure, continuity, ...) into the machine learning domain. Their basic premise is that low-level perceptual organization is distinct from semantic organization (categorization) -- a claim that isn't as cut-and-dry as the authors suggest. However, generally I think this is not an unreasonable connection to make.

**Weaknesses:**

Exposition of the problem domain could be greatly improved. Upon my initial read of the abstract and introduction, I was unsure (1) what problem domain we were in (vision), (2) what learning problem we are dealing with (semi-supervised learning / GCD), and (3) what architecture we are dealing with (the paper immediately talks about projections and backbones in the introduction without context). In all three cases, the paper jumps in with the assumption that the reader is aware of these things (like presenting to a collaborator rather than an appropriately knowledgable audience). I would recommend the authors dedicate more of the paper to exposition and motivation of the problem domain -- this would serve as a natural bridge to why certain ideas from cognitive science may be applicable.

Diving into the main method section, my main criticism of this paper is that it lacks a careful analysis of each inductive bias that is being introduced. The main evaluation is general improvement over standard benchmarks. For me to be convinced of the framing, I would need to see specific instances where gestalt principles enable correct identification of novel classes, where a non-gestalt baseline fails.

**Questions:**

Many of my questions were answered by reading Vaze 2022 -- which introduces GCD. To re-emphasize, the paper would greatly benefit from a better exposition of the problem domain. It should be a somewhat self-contained package.

---

### Meta-Review · Area_Chair_BZCs · 2026-01-05

**Summary:**

The main concerns from the reviewers are:

- Vague motivation and insight. The terminology is unnecessarily heavy and difficult to follow. The cognitive science framing, while appealing, lacks rigorous grounding.

- Lacks a careful analysis of each introduced inductive bias.

- Lacks hyperparameter choice.

- Marginal gains on coarse tasks.

- Incomplete, less than 9 pages.

- Missing critical experimental details.

**Reviewer Concerns:**

The authors did not respond. So I think the reviewers' concerns are not addressed, and I also shared many similar concerns about this work. I believe the paper should be revised and add more results and analses according to the advices from the reviewers.

**Reviewer Scores:**

I think the reviewers with negative scores would not change their scores. The only reviewer giving 6 may not increase the score and decrease the score, as the other reviewers have proposed several important problems of this submission, e.g., the vague motivation, insufficient results, and marginal improvement, in my opinion.

---

### Decision · Program_Chairs · 2026-01-26

Reject